# Exploring Healthcare Workers’ Knowledge and Perspectives on Behavioral Risk Factors Contributing to Non-Communicable Diseases: A Qualitative Study in Bushbuckridge, Ehlanzeni District, Mpumalanga Province, South Africa

**DOI:** 10.3390/ijerph22030343

**Published:** 2025-02-26

**Authors:** Thabo D. Pilusa, Cairo B. Ntimana, Mahlodi P. Maphakela, Eric Maimela

**Affiliations:** 1Department of Public Health, University of Limpopo, Sovenga St, Polokwane 0727, South Africa; tptpilusa@gmail.com (T.D.P.); eric.maimela77@gmail.com (E.M.); 2Dikgale Mamabolo Mothiba (DIMAMO) Population Health Research Centre, University of Limpopo, Sovenga St, Polokwane 0727, South Africa; 3Department of Student Affairs, University of Limpopo, Sovenga St, Polokwane 0727, South Africa; mahlodi.maphakela@ul.ac.za

**Keywords:** healthcare workers, non-communicable disease, knowledge, behavioral risk factors

## Abstract

Healthcare workers have been the backbone of information regarding behavioral risk factors and non-communicable diseases that have caused morbidity and mortality globally over the past decades. The study explores the knowledge of healthcare workers about behavioral risk factors contributing to non-communicable diseases. The study utilized a qualitative, explorative, and descriptive design. Data were collected through a semi-structured interview guide, involving eight healthcare workers from Bushbuckridge. Purposive sampling was used to select the participants. Healthcare workers were interviewed at their convenience, such as at lunch or as per their appointment time. Using thematic analysis, the researchers were able to systematically organize and interpret the data presented in the themes. Two themes and seven subthemes emerged regarding the knowledge of healthcare workers about behavioral risk factors contributing to non-communicable diseases. These themes, identified through deductive analysis, include behavioral risk factors and health system factors. The study found that a lack of seminars or training in the management of chronic disorders has left healthcare providers in Bushbuckridge with less information about behavioral risk factors related to non-communicable diseases. However, these findings reflect the perspectives of a small sample and require further investigation through broader qualitative and quantitative research to determine their generalizability and potential systemic implications.

## 1. Introduction

Non-communicable diseases (NCDs), including diabetes, cancer, cardiovascular disease, and chronic respiratory conditions, are responsible for over 70% of deaths worldwide, making them the leading global cause of mortality [1]. Many of these NCDs share common behavioral risk factors, such as smoking, unhealthy diets, lack of exercise, and excessive alcohol use that are largely modifiable [2,3,4]. These behaviors often lead to conditions such as obesity, high blood pressure, elevated cholesterol, and reduced physical activity, which contribute significantly to NCD development [5,6]. Despite considerable public health efforts, NCDs remain a challenge in all countries, especially in low- and middle-income nations where more than 75% of NCD-related deaths occur [1].

Comprehending the existing state and advancements at the national level is crucial for efficiently addressing NCDs and their primary risk factors [7]. Numerous practical and cost-effective strategies exist to mitigate the burden of NCDs now and in the future [7,8,9]. Monitoring key intervention activities at the national level enables global benchmarking, highlights progress against NCDs, and brings attention to areas needing further improvement [10,11].

The role of healthcare workers (HCWs) in preventing and managing NCDs is critical, as their knowledge, attitudes, and practices influence their ability to educate and support patients in adopting healthier lifestyles [12]. Previous studies have reported that healthcare workers’ knowledge, attitudes, and practices toward NCDs significantly affect their ability to educate and support patients in managing their conditions [13,14]. In rural settings, where healthcare systems are overstretched and health literacy among the general population is often low, the knowledge that HCWs possess about NCDs and their risk factors can determine the success of community-level interventions. In such contexts, HCWs play a pivotal role in delivering community-level interventions to reduce the prevalence of NCD risk factors. However, there is a notable gap in research on the extent of HCWs’ knowledge regarding NCD risk factors in rural South African communities [13]. Existing literature on HCWs’ knowledge and attitudes toward NCDs in South Africa has primarily focused on urban settings, leaving a critical knowledge gap regarding rural healthcare workers’ preparedness to address NCDs in resource-limited environments [15,16]. Hence, this study aimed to explore the knowledge of healthcare workers about behavioral risk factors contributing to non-communicable diseases. The findings of the present study will provide insights into potential gaps in their understanding, which can inform targeted educational programs, health policies, and interventions to curb the growing NCD epidemic in the region.

## 2. Materials and Methods

### 2.1. Research Design

This study employed a qualitative exploratory, descriptive, and phenomenological design. This flexible approach yields high-quality data that facilitates a deeper understanding of individuals’ lived experiences [17]. Additionally, the approach emphasized exploring a shared phenomenon or concept among the participants, one that the researchers were aware of but lacked detailed knowledge about. This shared concept pertained to the understanding healthcare workers have regarding behavioral risk factors that contribute to non-communicable diseases. The study was approved by the University of Limpopo Turflop Research Ethics Committee (Project number: TREC/1777/2023:PG) and the Mpumalanga Provincial Health Research and Ethics Committee (MPHREC) Reference Number: MP 202403 001.

### 2.2. Study Setting

The study was conducted in the Cottondale local area, which is in the Bushbuckridge sub-district of Mpumalanga Province. The local area consists of 6 fixed clinics and 1 district hospital (District Health Information System, 2017). The Ehlanzeni District Municipality is situated in the northeast of the Mpumalanga Province. The district comprises four local municipalities: Bushbuckridge, City of Mbombela, Nkomazi, and Thaba Chweu, with a total population of 1,743,182. The research setting for this study was healthcare facilities that are situated in the Bushbuckridge sub-district.

### 2.3. Study Population and Sampling

The target population of the study was healthcare workers (nurses) who were working in six selected healthcare facilities in Bushbuckridge, Ehlanzeni District, Mpumalanga Province. Purposive sampling was utilized for healthcare workers. Healthcare workers who have less than 2 years’ experience dealing with NCDs were excluded from the study, and those with experience of over 2 years were purposively selected to take part in the study. The inclusion of participants with diverse characteristics and the use of rich, illustrative quotations further strengthened the contextual relevance of the findings. Per institution, there were about 10 nurses who met the selection criteria.

#### Data Collection

Data were collected between 1 September 2024 and 30 October 2024 through qualitative face-to-face in-depth interviews using an interview guide. Arrangements for the date and time for data collection were made. Three principal authors (TDP, CBN, and EM) performed semi-structured one-on-one interviews in a private room with each participant. Three authors (TDP, CBN, and EM) conducted the interviews. Before the interviews were performed, the primary authors clarified the reason for conducting the interviews, the voluntary nature of participation, and the freedom to withdraw at any moment. And all participants signed the informed consent for before the enterviews were conducted. The interviews were conducted following the interview guide with questions asked (see Appendix A). Researchers recorded the sessions, which each lasted around 40 min, using audio recorders and notepads. Data collection continued until saturation was achieved with participant number eight, with multiple interviews per participant, when no new significant insights or themes emerged from new participants.

### 2.4. Data Analysis

The data analysis was conducted using Tesch’s open thematic approach, applied at multiple levels and based on textual data provided by patients [17]. Two authors, CBN and EM, managed the dataset, which included verbatim transcriptions of all recorded audio files and accompanying field notes. CBN and EM meticulously reviewed the transcripts to gain a thorough understanding of the content. The next step involved data coding, where meaningful segments were identified and assigned appropriate labels. These codes were then examined to identify and describe emerging themes. A collaborative meeting with the rest of the authors was held to refine and finalize these themes. The findings were reported as qualitative results, with all authors reaching a consensus on the themes and subthemes summarized in Figure 1. Tesch’s open thematic analysis enabled the authors to immerse themselves deeply in the dataset, allowing new insights to emerge naturally. The themes are presented in Figure 1.

### 2.5. Measures to Ensure Trustworthiness

To ensure the study’s accuracy and rigor, the researchers followed Lincoln and Guba’s four evaluation criteria [18], which offer a structured framework for evaluating the trustworthiness and robustness of qualitative research. Credibility was established through a member-checking process that incorporated feedback from three participants and the authors. Ample time was dedicated to data collection, interpretation, and iterative engagement with the data. Transferability was supported by providing detailed descriptions of the study’s setting, participants, and methods, enabling readers to evaluate the applicability of the findings to other contexts. Participants with varied characteristics were purposefully selected, and the findings were presented in a comprehensive and nuanced manner, enriched with relevant interview quotations. Confirmability was ensured through the creation of an audit trail, meticulously documenting study procedures, decisions, and any modifications, making the findings traceable and verifiable. To maintain consistency, the duration of data collection (interviews) was minimized, and all participants were asked the same set of questions. Dependability was achieved through transparent and detailed documentation of study procedures, data collection, analysis, and theme development. Additionally, the entire research process was reviewed by multiple researchers to validate its accuracy and rigor.

## 3. Results

In-depth interviews were conducted until data saturation was reached by eight [8] HCWs when no significant new insights or themes emerged from new participants. The majority of participants were female, all aged over 37 (Table 1). The Section 3 is structured around key themes and subthemes about the knowledge of healthcare workers about behavioral risk factors contributing to non-communicable diseases. These themes, identified through deductive analysis, include behavioral risk factors and health system analysis. These themes and their sub-themes are represented schematically in Figure 1.

### 3.1. Theme 1: Behavioral Risk Factors

This theme focuses on exploring the various behavioral factors that contribute to NCDs within the Bushbuckridge community. It involves studying lifestyle habits, such as diet, physical activity, smoking, and alcohol consumption, which have a significant impact on NCD prevalence. Understanding these factors can help in developing targeted interventions and preventive measures to address the root causes of NCDs within the Bushbuckridge community.

#### 3.1.1. Subtheme 1.1 Unhealthy Eating Habits and Poor Lifestyle Choices

The HCWs shared their personal experiences and observations regarding the possible causes of these health conditions. They mention these health conditions may be due to diet. Moreover, they further mentioned alcohol consumption being prevalent in the community, including among children, which raises concerns about the broader dietary and lifestyle habits contributing to health issues. This is what they had to say:


*“…mostly I think here could be diet and “yah”, I think I could diet because they eat everything and I think alcohol because here everyone drinks from grandmother, mother even children…” *
(HCW 4, female, aged 56)


*Additionally, alcohol consumption appears to be prevalent across all age groups, raising further concerns about overall dietary and lifestyle habits.*


#### 3.1.2. Subtheme 1.2 Mental Health Problems and Stress

HCWs shed light on the pervasive issue of mental health problems and stress, particularly among females. They highlighted a gendered aspect, suggesting that females tend to internalize stress more than males. Participants also expanded by indicating that stress often manifests as hypertension and is exacerbated by lifestyle factors. The feelings of the participants are shared in the following extracts:


*“Err” when I see to the side of female is stress, they are having stress, they think too much than males…” *
(HCW 3, female, aged, 37)


*“You mean hypertension only” “ooh” both, it is lifestyle, you find that a person has stress and he/she does not have a person to share with like if a person has lost a child or husband through death, this person thinks too much…” *
(HCW 5, female, aged 45)

#### 3.1.3. Subtheme 1.3 Understanding of Management of Chronic Conditions

The HCWs in this study highlighted a reliance on visiting doctors for education and training regarding the management of hypertension and diabetes, with limited formal workshops or specific training opportunities available to healthcare workers. The following assertions illustrate these findings:


*… “Err, in many cases visiting Doctors are the one teaching us in relation to hypertension and Diabetes prevention and management, but no workshop was done, even if the workshop is done, the teaching is not mainly on Diabetes and hypertension.” *
(HCW 1, male, aged 37)


*… “Mm, as for me I have not attended any workshop about prevention and management of hypertension and diabetes. I only have little information from the previous hospital where I was working before, I came here. Apart from training we depend on visiting doctors for in-service education regarding the management of hypertension and diabetes when they visit our facility every Tuesday.” *
(HCW 2, female, aged 37)


*… “No, as for me I have not attended any training regarding behavioral risk factors for hypertension and Diabetes, I just know how to manager patient because I just read the books and working alongside with doctors, we hear them, and they teach.” *
(HCW 3, female, aged 37)

### 3.2. Theme 2: Health System Factors

This theme revolved around investigating the healthcare system factors that influence the occurrence and management of NCDs in Bushbuckridge. It includes assessing aspects such as healthcare infrastructure, access to healthcare services, availability of medical personnel, quality of care, and health education programs. Identifying strengths and weaknesses within the health system can inform strategies for improving NCD prevention, diagnosis, and treatment outcomes.

#### 3.2.1. Sub-Theme 2.1: Availability of Screening and Monitoring Tools


*The participants highlighted that they lack screening and monitoring equipment to assist them in caring for the patients that are having chronic conditions. The lack of working equipment can hinder the quality of care of patients with hypertension and Diabetes.*



*… “Err, when coming to Hypertension and Diabetes, our health program is integrated with other stakeholders such as community health workers, we work together with community healthcare workers who help us to trace patients who defaulted their treatment and to add on that they go door to door with a machine that they use to test sugar and if they detect abnormal sugar level they refer them to the clinic for further management however currently they do not have a machine to test hypertension.” *
(HCW 1, male, aged 37)


*… “Home base carers, we work with them in this way, when they do home visits, they give health education about lifestyle in the villages, and we also offered them machines to check blood sugar within the villages, but we did not give them a machine to check hypertension…” *
(HCW 3, female, aged 37)

#### 3.2.2. Sub-Theme 2.2: Health Education Provision Platforms

In this study, the HCWs indicated that they actively engaged in health education activities to promote healthy behaviors among patients. Participants highlighted that they conducted health education sessions every morning before beginning their clinical duties, often addressing patients who were waiting for consultations. The following extracts reflect their experiences:

*… “Mm”, we give them health education, teaching them about eating of healthy diet, encouraging them to exercise, avoid stressful situation although it is not easy, and we also give them treatment and encourage them to take it as prescribed”…*.(HCW 2, female, aged 44)


*… “Ok, err that one is through health education, each morning before we start working…in all the patient whether children or whatsoever, we teach them about hypertension, diabetes, HIV, we teach them, I think that one will assist”…*
(HCW 6, male, aged 36)


*… “Err, in the morning we give health education and it depend on what health education is given that day, when you come to the consultation room I give one on one health education, all the information is given, if the BP is OK we do praise the patient wow this BP is nice you should keep it like that and we give information to prevent, yah to prevent. There is a specific day for health education not every day” …*
(HCW 4, females, aged 45)

#### 3.2.3. Sub-Theme 2.3: Collaboration Between Healthcare Workers and Other Stakeholders

In this study, the HCWs emphasized the importance of a team approach to managing chronic illnesses between community workers and healthcare professionals, with each group providing specialized roles and support to guarantee patients receive full treatment in the community.


*… “Err, when coming to Hypertension and Diabetes, our health program is integrated with other stakeholders such as community health workers, we work together with community healthcare workers who help us to trace patients who defaulted their treatment and to add on that they go door to door with a machine that they use to test sugar and if they detect abnormal sugar level they refer them to the clinic for further management however currently they do not have a machine to test hypertension”…*
(HCW 1, male, aged 37)


*…“Err, we call them CHW, they work there in the field, they check chronic patient in their home, they bath those who are unable to bath, they collect medication for patients who are unable to come and collect and some here as you saw them on the table and since we do not have help desk nurses, they work as our help desk nurses and they also give patient sputum bottles”… *
(HCW 8, male, aged 53)

#### 3.2.4. Sub-Theme 2.4: Utilization of the Service

The HCWs alluded to how the community members utilize healthcare services and recognize the effectiveness of the services, especially in managing chronic conditions like hypertension and diabetes.


*… “Err, the community do utilize our services and we work together with them however in most cases they do not comply to our instructions such as return dates given to them but some of the community members do comply to our instructions”…*
(HCW 1, male, aged 37)


*… “Mmm, according to me I see the community using our services effectively because when they are sick, they come to our clinic for consultation”…… *
(HCW 2, female, aged 44)


*… “Yes, they use our service in place effective just because they hypertension is controlled, diabetes is controlled since we have doctor “For” visiting and teach the patient about lifestyle”… *
(HCW 3, female, aged 37)

## 4. Discussion

The study aimed to explore the knowledge of healthcare workers about behavioral risk factors contributing to NCDs. HCWs shared personal experiences and insights that revealed the complex interconnections between health conditions and broader lifestyle practices. They pointed out that these conditions are influenced not only by individual dietary choices but also by society-wide dietary and lifestyle standards. For example, the prevalence of alcohol consumption in the community, even among children, raises concerns about the influence of collective habits on health issues. In agreement with the findings of the present study, Jayedi et al. [19] emphasized the critical role of healthy eating habits in preventing and managing type 2 diabetes and hypertension, underscoring the detrimental impact of poor dietary patterns on these conditions.

Additionally, HCWs observed a gendered dimension of stress, noting that women tend to internalize stress more than men, which aligns with societal expectations and gender roles. Stress was highlighted as a significant contributor to hypertension, exacerbated by lifestyle factors and a lack of support systems. The HCWs emphasized the importance of social support in managing stress and mental health, suggesting that resilience and coping skills are vital for promoting psychological well-being. These observations are consistent with findings by Gorman et al. [20], who explored stress, coping mechanisms, and depression across genders, identifying potential areas for targeted intervention and support.

The study also underscored that HCWs relied on doctors for education and training in managing hypertension and diabetes, as there are limited formal workshops or specialized training opportunities. In agreement with the findings of the present study, Gouglas et al. [21] noted that primary healthcare practitioners in South Africa have insufficient knowledge about hypertension and the corresponding national hypertension guidelines. The realization of the above findings accentuates the need to invest in ongoing education, training programs, and knowledge-sharing initiatives [21]. Healthcare organizations can support workers in providing high-quality care and improving outcomes for patients with chronic conditions adherence [22,23].

A notable issue raised was the lack of screening and monitoring equipment available to HCWs, hindering their ability to care for patients with chronic conditions like hypertension and diabetes effectively. From the above findings, it has been identified that there is a gap in healthcare infrastructure, where essential equipment necessary for diagnosing and managing chronic conditions is lacking, potentially limiting access to appropriate care for patients. These reports were confirmed by a study carried out by Prothero et al. [24], whose findings were a significant barrier to effective diabetes management in the form of inadequate access to essential screening and monitoring tools. This limitation can impede the timely detection of diabetes-related complications and hinder the adjustment of treatment plans. This calls for a need for expanding access to training programs by establishing more clinic mobiles, community health initiatives, and telemedicine services to ensure regular and widespread NCD screening, particularly in underserved areas.

The HCWs highlighted their efforts to engage patients with NCDs on various health aspects, including nutrition, exercise, stress management, and disease prevention. They also emphasized the significance of personalized health education, tailored to individual patient needs and health conditions. In agreement with the present study, Melariri and colleagues [25] reported similar findings, where they highlight the importance of regular health education sessions, conducted typically in the morning before commencing work, wherein HCWs educate patients on various aspects of health, including nutrition, exercise, stress management, and disease prevention. Furthermore, it emphasizes the significance of personalized health education, tailored to individual patient needs and health conditions [25,26,27]. Positive reinforcement, such as praising patients for maintaining healthy blood pressure levels, was also identified as a valuable strategy in promoting patient adherence to health recommendations [28,29].

Lastly, the HCWs emphasized the importance of a team approach to managing chronic illnesses between community workers and healthcare professionals, with each group providing specialized roles and support to guarantee patients receive full treatment in the community. The findings of the present study corroborate a systematic review by Viswanathan et al. [30], which examined the role of community health workers in improving patient outcomes in rural chronic disease settings in the United States, highlighting their effectiveness in providing support and improving access to care for patients with chronic illnesses.

### Study Limitations

The study used a small sample size, which may limit the ability to generalize the results to healthcare workers in Mpumalanga or comparable settings. Additionally, the sampling method used to select participants may introduce potential bias. Nevertheless, the findings offer insights into the understanding of the knowledge of healthcare workers about behavioral risk factors contributing to non-communicable diseases in rural South Africa.

## 5. Conclusions

The interviewed healthcare providers in Bushbuckridge perceive a gap in their knowledge about behavioral risk factors for non-communicable diseases, which they attribute to limited access to seminars and training on chronic disease management. They emphasized that these diseases are influenced not only by individual dietary choices but also by broader societal dietary and lifestyle norms. Additionally, some participants highlighted a gendered dimension of stress, suggesting that women may internalize stress more due to societal expectations and gender roles. Stress was commonly linked to hypertension, exacerbated by lifestyle factors and a lack of support systems. However, these findings reflect the perspectives of a small sample and require further investigation through broader qualitative and quantitative research to determine their generalizability and potential systemic implications.

Healthcare workers underscored the critical role of social support in managing stress and mental health. They actively engage patients with chronic conditions on various health topics such as nutrition, exercise, stress management, and disease prevention. Personalized health education tailored to the specific needs and conditions of patients was also emphasized. Furthermore, they highlighted the value of a collaborative team approach in managing chronic illnesses, where community workers and healthcare professionals bring their specialized skills together to ensure comprehensive patient care within the community.

Based on the findings of the present study, it is essential to increase healthcare professionals’ awareness of behavioral risk factors for non-communicable diseases by offering training on managing these conditions in facilities aligned with WHO policies, protocols, and guidelines. Tackling behavioral risk factors calls for a holistic strategy that combines health education, lifestyle changes, and robust support systems. Additionally, continuous education and training programs are crucial for empowering healthcare workers and enhancing long-term patient outcomes.

## Figures and Tables

**Figure 1 ijerph-22-00343-f001:**
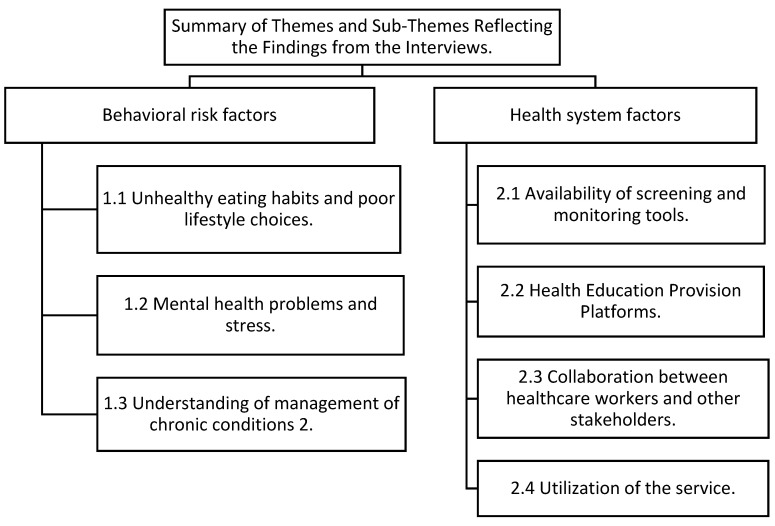
Summary of themes and sub-themes reflecting the findings from the interviews.

**Table 1 ijerph-22-00343-t001:** Demographic characteristics of participants.

Pseudo Names	Age	Gender	Level of Education	Marital Status
HCW 1	37	Male	Tertiary	Married
HCW 2	44	Female	Tertiary	Married
HCW 3	37	Female	Tertiary	Married
HCW 4	56	Female	Tertiary	Single
HCW 5	43	Female	Tertiary	Widowed
HCW 6	36	Male	Tertiary	Married
HCW 7	53	Male	Tertiary	Married
HCW 8	53	Male	Tertiary	Single

## Data Availability

The data presented in this study are available on request from the corresponding author. The data are not publicly available due to privacy or ethical restrictions.

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
