# Peer review of "Exploring Healthcare Workers’ Knowledge and Perspectives on Behavioral Risk Factors Contributing to Non-Communicable Diseases: A Qualitative Study in Bushbuckridge, Ehlanzeni District, Mpumalanga Province, South Africa"

_ijerph, 2025, doi:10.3390/ijerph22030343_

Round 1
Reviewer 1 Report
Comments and Suggestions for Authors
Methods: definition of healthcare workers should be explicit, including their education, experience and are of work etc.
The process of selection should be detiled for the readers to understand how the sample was taken...only 8 respondents from 6 facilities seem too low, even for a qual study...whether respondents categorised by age, sex , experience, SES etc, which have bearings on their KAP related to the NCD knowledge and management
A Table to show themes, sub-themes, relevant codes etc. is needed for readers to grasp the strength of the evidence generated.
not HS analysis but should be HS factors...details of the system is also needed to contextualsie the findings...
Comments on the Quality of English Language
problems with grammar and style...too many long sentences, confusing sentences, repetition etc.
Needs thorough editiing by a professional editor!
Author Response
Dear reviewer
Thank you for the comments
Please find the table of corrections attached

Reviewer 2 Report
Comments and Suggestions for Authors
Overall, this article is well presented and addressing an important issue of behavioral risk factors contributing to non-communicable diseases. The manuscript is written well through introduction, methods, results, discussion, and conclusions. However, there’s only minor comments as demonstrated below
· In the study period, add date, From …….to……… (M/D/Y)
· Better to add Flowchart for demonstrating themes, and subthemes headings
· Line 225: Sub-theme 2.7: the authors shift from subtheme 2.3 to 2.7 where’s subtheme 2.4, 2.5,2.6.

Author Response
Dear Reviewer
Thank you for the comments
Please find the table of corrections attached

Reviewer 3 Report
Comments and Suggestions for Authors
The intent of this research was clearly described. The paper referred to the use of phenomenology however this methodology/method was not described beyond the distillation of themes and subthemes from the transcripts.
The results achieved the aim stated and provided insights highlighting knowledge concerns of HCWs and the inadequacy of resources to support their work. In addition, the impact of alcohol on the community including children is concerning.
The recommendations arising from the study were realistic and reflected the outcomes. The limitations of the study were thoughtful and realistic.
Author Response
Dear reviewer
Thanks for the comments
Please find the attached document of corrections

Reviewer 4 Report
Comments and Suggestions for Authors
Dear authors,
Although the study addresses a critical issue in Public Health, the manuscript presents numerous weaknesses, as identified below:
1. Title and Summary Section:
· The title should be more specific about the qualitative nature of the study.
· The abstract should provide more details about the findings and their implications, as well as detail the methodological approach and sample size.
2. Introduction Section:
· The literature review should be more comprehensive, and include more recent studies and a broader context.
· The introduction should be more specific about the objectives of the study, in addition to specifying a clear hypothesis or research question.
3. Materials and Methods Section:
· The sample size of eight participants may have limited the generalizability of the results.
· Possible biases in the data collection process were not discussed in detail.
· Ethical considerations and approvals were not adequately addressed.
· The interview guide and data saturation process should have been described in more detail.
4. Results Section:
· The integration of quotes and themes should be improved for greater coherence.
· The subthemes could be presented in more detail.
· The quotes could be better contextualized for better understanding.
5. Discussion Section:
· The literature review should be more comprehensive, including a wider range of studies. It lacks depth in discussing the findings and implications of the cited studies.
· Some of the references are out of date.
· Some interpretations in the discussion are speculative and not well supported by the data.
· The limitations of the study were not discussed in detail, especially the sample size and possible biases.
6. Conclusion Section:
· The conclusion should be more specific about the study's findings and their implications, and should be described more clearly and concisely.
I wish you success with the manuscript.
Author Response
Dear reviewer. Thank you for the comments
please find the table of amendments attached

Reviewer 5 Report
Comments and Suggestions for Authors
The focus of this manuscript is needed to address the need for prevention and management of chronic disease. This is a relevant topic. However, the methodology is lacking. Much information is missing. How did the authors assure fidelity and trustworthiness? What were the steps taken? A brief overview of the methodology used should be provided. It is hard to understand how the authors can to the conclusions made without knowledge of the interview guide. What questions were asked? Also, how did saturation come about? How did you know when saturation was achieved? Your conclusions don't match your methodology. See comments on attached. Additionally, there are some references that are too old. It would be best to have references within 5 years.

English overall was ok, but there are grammar issues. The direct quotes from the participants are difficult to understand, and I am not sure the intent of their comments translated into English
Author Response
Dear reviewer
Thanks for the comments
Please find the attached document of amendments

Round 2
Reviewer 5 Report
Comments and Suggestions for Authors
This paper has addressed several of my previous concerns. However, there needs to be improved description of the sampling and inclusion criteria, as well as how trustworthiness was guaranteed wit the interview process. Also, the abstract needs to match what was written in the paper. Please see the comments about your conclusions. Additionally, there are some grammatical errors and misspellings that should be corrected.

Some misspelled words and grammar errors
Author Response
Dear Reviewer
Thanks for the comments
Please find the attached table of corrections
